# Prevalence and associated factors of visual impairment among adult diabetic patients visiting Adare General Hospital, Hawassa, South Ethiopia, 2022

Henok Biruk Alemayehu[1], Melkamu Temeselew Tegegn[2], Mikias Mered Tilahun🔟[2]*

1 Department of Ophthalmology and Optometry, Collage of Medicine and Health Sciences, Hawassa University, Hawassa, Ethiopia, 2 Department of Optometry, School of Medicine, University of Gondar, Comprehensive Specialized Hospital, Gondar, Ethiopia

* mikiserke123@gmail.com

## Abstract

### Background

The increased prevalence of visual impairment among diabetic patients has become a major public health problem. However, there was limited information on the extent of visual impairment among diabetic patients in our country, and there was no study in the study area. Providing updated data regarding this area is critical for the prevention of visual impairment among diabetic patients.

### Purpose

The study aimed to assess the prevalence and associated factors of visual impairment among adult diabetic patients visiting Adare General Hospital, Hawassa, South Ethiopia, 2022.

### Methods

A hospital-based cross-sectional study was conducted on adult diabetic patients from May 30 to July 15, 2022, at Adare General Hospital, Hawassa, South Ethiopia. A systematic random sampling method was used to select 398 study participants. Data was collected through a face-to-face interview, a medical chart review, and an ocular examination. A binary logistic regression was performed to identify potential risk factors for visual impairment and their strength of association was expressed using an adjusted odds ratio with a 95% confidence interval. Variables with a P-value of < 0.05 were considered statistically significant.

### Result

In this study, a total of 391 participants were involved, with a response rate of 98.2%. The prevalence of visual impairment was 28.6% (95% CI: 24.6–33.0). Age $\geq$ 60 years (AOR = 4.03, 95% CI: 1.72, 10.71), poor physical exercise (AOR = 3.26, 95% CI: 1.62, 6.53), poor

**Data Availability Statement:** All relevant data are available on figshare at: https://doi.org/10.6084/m9.figshare.21229829.

**Funding:** The author(s) received no specific funding for this work.

**Competing interests:** The authors have declared that no competing interests exist.

glycemic control (AOR = 4.34, 95% CI: 2.26, 8.34), history of eye examination (AOR = 2.94, 95% CI: 1.50, 5.76), duration of diabetes ≥ 9 years (AOR = 4.78, 95% CI: 2.11, 10.83) and diabetic peripheral neuropathy (AOR = 3.01, 95% CI: 1.21, 7.50) were positively associated with visual impairment.

## Conclusion

The study found a high prevalence of visual impairment among adult diabetic patients. Older age, longer duration of diabetes, poor physical exercise, poor glycemic control, history of eye examination, and diabetic peripheral neuropathy were significantly associated with visual impairment. Thus, regular physical activity, good control of glucose levels, and regular eye exams were recommended for all diabetic patients.

## Introduction

Diabetes mellitus is a group of metabolic disorder characterized by high blood glucose levels that results from defects in insulin secretion, action, or both [1]. Approximately 537 million people are living with diabetes globally, of whom 24 million people are living in Africa [2]. Moreover, in Ethiopia, the prevalence of diabetes mellitus ranges from 0.34% to 12.4% [3–5]. Visual impairment is defined as the loss of functionality of the eye(s) or visual systems, and it is manifested by decreased visual acuity, visual field loss, visual distortion, or perception problems [6]. Visual impairment is more common in people with diabetes than in people without diabetes [7, 8].

Globally, about 2.2 billion people have a near or distance visual impairment, of whom 3.9 million are visually impaired due to diabetic retinopathy [9]. In Africa, the prevalence of visual impairment among diabetic patients ranges from 17.1% to 78.25% [10–13]. Studies done in Dessie and Gondar, Ethiopia showed that the prevalence of visual impairment among diabetic patients was 37.58% and 70.06%, respectively [14, 15].

Visual impairment among diabetic patients can increase the unemployment rate, reduce productivity, increase medical expenses, reduce the performance of daily living activities and reduce social participation. Those conditions lead an individual with diabetes to have a reduced quality of life [9, 16].

Studies indicate that the main factors that are associated with visual impairment among diabetic patients are poor glycemic control level, poor physical exercise, older age, long duration of diabetes, and type of treatment [15, 17, 18]. Evidence showed that controlling blood glucose levels, having a regular eye exam, and undergoing early laser photocoagulation have been used to reduce the burden of visual impairment among diabetic patients [19–21].

The increased prevalence of visual impairment among diabetic patients has become a major public health problem in both developing and developed countries that requires significant attention [22–24]. Despite this, there is limited evidence on the magnitude of visual impairment and associated factors in Ethiopia, particularly in the study area. Thus, the main objective of this study was to assess the prevalence and associated factors of visual impairment among diabetic patients visiting Adare General Hospital, Hawassa, South Ethiopia.

This study will provide valuable information for the planning and prioritization of health care programs to early detect and treat visual impairment among people with diabetes in the region.

## Methods and materials

### Study design, area and period

A hospital-based cross-sectional study was conducted at Adare General Hospital from May 30 to July 15, 2022. Adare General Hospital is located in Hawassa, the capital city of Sidama Regional State, which is located 275 km south of Addis Ababa. According to the hospital's planning and information office, Adare General Hospital provides both preventive and curative health services, including eye care services, for about three million people in the region. Besides, it has the largest diabetic center, serving more than 600 diabetic patients per month on five working days per week. Internists, general practitioners, and nurses were involved in the clinical care of diabetic patients. The ophthalmology department in Adare General Hospital is providing comprehensive eye care service for the surrounding community through 1 ophthalmologist, 5 optometrists, and 1 ophthalmic nurse.

### Source and study population

All adult, aged $\geq$ 18 years' patients with type I or type II diabetes who visited the diabetic clinic at Adare General Hospital were a source population, and All adult, aged $\geq$ 18 years' patients with type I or type II diabetes who visited the diabetic clinic at Adare General Hospital diabetic clinic during the data collection time were eligible to participated in the study. However, all adult, aged $\geq$ 18 years' patients with type I or type II diabetes who were admitted due to critical illness, as a result, were unable to be examined with a slit lamp, and those with less than 3 months of follow-up were exclude from the study.

### Sample size determination and sampling procedure

Sample size was determined using a single population proportion formula by considering the expected proportion of visual impairment among diabetic patients from a previous study in Dessie, Ethiopia (37.5%) [15], 95% of confidence level, and 5% degree of precision. So, the calculated sample size was 361, and then adds 10% of non–response rate, the final calculated sample size was 391. A systematic random sampling technique with an interval of 2 was applied to select the study participants. A single number was taken using a lottery method to select the first study participant and continued with every $K^{th}$ interval. An interval was calculated by dividing the expected number of diabetic patients who came to the diabetic clinic during the data collection period by the required sample size (K = N/n, N = 954, n = 398).

### Operational definitions

**Visual impairment.** Was defined as a presenting visual acuity less than 6/12 in the better eye based on the International Classification of Diseases 11[th] definition of visual impairment [25].

**Physical exercise.** A person who experienced regular exercise for less than 150 min (3–5 days) per week was considered to have poor regular exercise; otherwise, it was considered to have good regular exercise [26]

**Level of glycemic control.** It was classified as good when recorded fast blood sugar (FBS) was below 152 mg/dl and poor when FBS was 152 mg/dl and above [27].

**Body mass index.** (kg/m2) was calculated as weight (kg) divided by height in square meters (m2) and was graded according to the World health organization classification. A BMI of <18.5 was underweight, a BMI of 18.5–24.9 kg/m2 was normal, a BMI of 25–29.9 kg/m2 was overweight, and a BMI of $\geq$ 30 kg/m2 was obese [28].

**Age.** Was categorized as 18–40 years, 41–59 years, and 60–80 years based on a study done in Dessie, Ethiopia [15].

**Marital status.** Was categorized as currently married and currently single (single, divorced, and widowed).

## Data collection procedures and personnel

An interviewer-administered an Amharic version of the pre-tested structured questionnaire; meter for height; balance meter for weight; Snellen visual acuity chart; a pinhole, a slit lamp, and a 90D volk lens for anterior and posterior segment examination were used for data collection. Furthermore, clinical data such as fasting blood glucose level, duration of DM, and systemic co-morbidities were collected from each patient's medical chart. Besides, peripheral neuropathy was diagnosed based on a person's medical history and by performed physical examination such as inspection of feet, vibration perception, ankle reflex test, and monofilament test which was done by senior nurse.

Four well-trained data collectors (two nurses and two optometrists) participated in the data collection. The nurses conducted a face-to-face interview using a pre-tested structured questionnaire, which consists of information on socio-demographic and behavioral data. The clinical data, height, and weight of the study participants were also measured by nurses. After completing the interview, all study participants received a comprehensive ocular examination, which was done by optometrists. Visual acuity was tested in each eye for each study participants using a Snellen visual acuity chart at a distance of 6 meters. When the participants could not see a letter at 1 meter, then count finger, hand motion, light perception, and no light perception was used to measure visual acuity. To determine whether the reduction of visual acuity was caused by refractive error or not, pinhole visual acuity was performed on each study subject whose distance visual acuity was less than 6/9. Following this, intraocular pressure was measured with a tonopen by the optometrist. The optometrists examined the anterior segment with a slit lamp biomicroscope. Moreover, posterior segment examination with a dilated pupil through 1% Tropicamide was conducted using a slit lamp and a 90D volk lens. For eyes with two or more diseases that may have caused visual impairment, the cause that had the presumed greatest impact on visual impairment was regarded as the primary diagnosis. Inter examiner reliability was checked and the value of Cohen's kappa statistics was 0.99 [22].

## Data quality control

Data quality was controlled by using a pre-tested Amharic version structured questionnaire, and the pre-test was done on 5% of the sample size at Hawassa University Specialized Comprehensive Hospital. Data quality was maintained by giving training for data collectors on how to collect the data and supervision was conducted during data collection. Moreover, the collected data was checked for completeness to ensure data quality at the end of the day.

## Data processing and analysis

After checking the completeness and consistency of the data, the data was coded, entered, and cleaned using EpiData version 3.1, and then was exported to Statistical Package for Social Science (SPSS) version 25 for analysis. Descriptive statistics such as proportion, frequency, and summary statistics (median and interquartile range), were used to summarize the descriptive part of the study. A binary logistic regression model was fitted to identify the possible risk factors associated with visual impairment. Multi-collinearity was checked using Variance Inflation Factor and Tolerance. Variables with a P-value of less than 0.2 in the bivariate analysis were entered into a multivariable logistic regression. The fitness of the model was checked by

using Hosmer and Lemeshow goodness of fit. The strength of association between independent and dependent variables was expressed using an adjusted odds ratio with a 95% confidence interval. A variable with a P-value of less than 0.05 was considered statistically significant.

### Ethical considerations

Before conducting the study, ethical approval was obtained from the University of Gondar, College of Medicine and Health Sciences, School of Medicine, Ethical Review Committee, and a formal permission letter was also obtained from the Adare General Hospital medical director. The aim of the study was properly explained to each study subject and then written informed consent was obtained from each study participant. All the study participants were informed about their right to withdraw from the study at any time during the interview and examination. The confidentiality of the study participants was secured by avoiding any personal identifiers from the data collection tools, and the data was locked. Patients were linked to the eye clinic when necessary for additional care and follow-up.

## Result

### Socio-demographic characteristics of the study participants

A total of 391 participants were involved in the study, with a response rate of 98.24%. The median age of the participants was 49 years with (IQR: 40–58). Out of 391 study participants, 197 (50.4%) were male. Most of the participants (61.9%) were urban dwellers (Table 1).

### Clinical, behavioral, and systemic comorbidities

The median duration of diabetes was 6 years (IQR: 3–9). The median value for FBS was 160 mg/dl (IQR: 130–179 mg/dl). Sixty-eight percent of participants had never visited an eye care clinic in their lifetime. Among the total study participants, 60 (5.3%) had a history of hypertension (Table 2).

### Prevalence of visual impairment

This study revealed that the prevalence of visual impairment was 28.6% (95% CI: 24.6, 33.0%), of whom 5.10% mild, 16.6% moderate, and 6.10% had severe visual impairment.

### Ocular abnormalities among study participants

Out of 391 study participants, 60 (15.4%) and 38 (9.7%) had presented with diabetic retinopathy and refractive error, respectively. In this study, the major causes of visual impairment were diabetic retinopathy (36.6%), followed by cataract (26.8%), and refractive error (16.1%) (Table 3).

### Factors associated with visual impairment

By using a bivariable analysis, age, residency, monthly income, physical exercise, history of eye examination, DM duration, BMI > 25, glycemic control, peripheral neuropathy, and hypertension were independently associated with visual impairment. However, in multivariable analysis, age, physical exercise, history of eye examination, duration of DM, glycemic control, and peripheral neuropathy remained significantly associated with visual impairment.

Participants aged 60–80 years were 4.30 times (AOR = 4.03, 95% CI: 1.72, 10.71) more likely to develop visual impairment than those aged 20–40 years. Poor physical exercise habits

**Table 1. Socio-demographic characteristics of diabetic patients visiting Adare General Hospital, Hawassa, South Ethiopia, 2022 (n = 391).**

| Variables | Categories | Frequency | Percent |
|---|---|---|---|
| **Age (year)** | 20–40 | 100 | 25.6 |
| | 41–59 | 198 | 50.6 |
| | 60–80 | 93 | 23.8 |
| **Sex** | Male | 197 | 50.4 |
| | Female | 194 | 49.6 |
| **Residency** | Urban | 242 | 61.9 |
| | Rural | 149 | 38.1 |
| **Marital status** | Currently married | 290 | 74.2 |
| | Currently single | 101 | 25.8 |
| **Educational status** | No formal education | 151 | 38.6 |
| | Primary school | 69 | 17.7 |
| | Secondary school | 74 | 18.9 |
| | College and University | 97 | 24.8 |
| **Occupational status** | Employed | 108 | 27.6 |
| | Merchant | 76 | 19.4 |
| | Farmer | 52 | 13.3 |
| | Daily laborer | 25 | 6.4 |
| | Housewife | 85 | 21.7 |
| | Other* | 45 | 11.6 |
| **Monthly income (Ethiopian birr)** | ≤4000 | 106 | 27.1 |
| | 4001–6000 | 110 | 28.1 |
| | 6001–7500 | 75 | 19.2 |
| | >7500 | 100 | 25.6 |
| **Health insurance** | Yes | 177 | 45.3 |
| | No | 214 | 54.7 |
| **Living status** | Alone | 64 | 16.4 |
| | With family member | 327 | 83.6 |

Other *: include retired and no job n = sample size, family monthly income was classified based on interquartile

increased the risk of visual impairment by 3.26 times (AOR = 3.26, 95% CI: 1.62, 6.53) compared to good physical exercise habits. The odds of visual impairment for those participants with a duration of diabetes since diagnosis of 7–9 years were 4.78 times (AOR = 4.78, 95% CI: 2.11, 10.83) more than those participants with a duration of diabetes ≤ 3 years. Participants who had poor glycemic control were 4.34 times (AOR = 4.34, 95% CI: 2.26, 8.34) more likely to be visually impaired than participants with good glycemic control. In comparison to people who had eye exams, those who had none were 2.94 times (AOR = 2.94, 95% CI: 1.50, 5.76) more likely to be visually impaired. Participants with peripheral neuropathy were 3.01 times (AOR = 3.01, 95% CI: 1.21, 7.50) more vulnerable to developing visual impairment as compared to participants without peripheral neuropathy (Table 4).

## Discussion

In the present study, the prevalence of visual impairment was 28.6% (95% CI: 24.6%, 33.0%). This result was in line with the study done in Cameroon, 29.7% [11].

On the other hand, the finding of this study was lower than the previous studies conducted in Dessie Ethiopia 37.57% [15], Gondar Ethiopia 70.06% [14], and Yemen 76.5% [29]. This

**Table 2. Clinical, behavioral, and systemic comorbidity of diabetic patients visiting Adare General Hospital, Hawassa, South Ethiopia, 2022 (n = 391).**

| Variables | Categories | Frequency | Percent |
|---|---|---|---|
| **Type of DM** | Type I | 117 | 29.9 |
| | Type II | 274 | 70.1 |
| **Duration of DM since diagnosis (years)** | ≤3 | 100 | 25.6 |
| | 4–6 | 130 | 33.2 |
| | 7–9 | 78 | 19.9 |
| | >9 | 83 | 21.3 |
| **Glycemic control** | Good | 154 | 39.4 |
| | Poor | 237 | 60.6 |
| **History of an eye examination** | Yes | 125 | 32 |
| | No | 266 | 68 |
| **BMI(kg/m$^2$)** | Normal | 290 | 74.2 |
| | Underweight | 38 | 9.7 |
| | Overweight and obese | 63 | 16.1 |
| **Physical exercise** | Poor | 269 | 68.8 |
| | Good | 122 | 31.2 |
| **Treatment mode** | Tablet | 193 | 49.4 |
| | Insulin | 159 | 40.6 |
| | Combined | 39 | 10.0 |
| **History of ocular surgery** | Yes | 8 | 2.10 |
| | No | 383 | 97.9 |
| **Systemic comorbidities** | | | |
| **Hypertension** | Yes | 60 | 5.3 |
| | No | 331 | 84.7 |
| **Peripheral neuropathy** | Yes | 31 | 7.9 |
| | No | 360 | 92.1 |
| **Heart disease** | Yes | 9 | 2.3 |
| | No | 382 | 97.7 |
| **Anemia and nephropathy** | Yes | 7 | 1.8 |
| | No | 384 | 98.2 |

Duration of DM was classified by interquartile

**Table 3. Ocular abnormalities among adult diabetic patients visiting Adare General Hospital, Hawassa, South Ethiopia, 2022.**

| Ocular abnormality | Number of visual impairment (%) | Overall (%) |
|---|---|---|
| Diabetic retinopathy | 41(36.6) | 60(15.4) |
| Refractive error | 18(16.1) | 38(9.7) |
| Cataract | 30(26.8) | 37(9.5) |
| Glaucoma | 14(12.5) | 24(6.1) |
| Other* | 9(8.0) | 13(3.3) |
| Normal | 0(0.0) | 219(56) |
| Total | 112(100) | 391(100) |

Other*: Age related macular degeneration and corneal opacity

**Table 4. Factors associated with visual impairment among adult diabetic patients visiting Adare General Hospital, Hawassa, South Ethiopia, 2022 ($n$ = 391).**

| Variable | Visual impairment | | COR (95% of CI) | AOR (95% of CI) | P-value |
|---|---|---|---|---|---|
| | Yes | No | | | |
| **Age (year)** | | | | | |
| 20–40 | 12 | 88 | 1.00 | 1.00 | |
| 41–59 | 62 | 136 | 3.34(1.70, 6.55) | 2.24(0.99, 5.09) | 0.53 |
| 60–80 | 38 | 55 | 5.06(2.43, 10.52) | 4.30(1.72, 10.71) | 0.002 |
| **Residency** | | | | | 0.80 |
| Urban | 60 | 182 | 1.00 | 1.00 | |
| Rural | 52 | 97 | 1.62(1.04, 2.59) | 1.07(0.60, 1.91) | |
| **Monthly income (Ethiopian Birr)** | | | | | 0.367 |
| ≤4000 | 39 | 67 | 2.06(1.11, 3.82) | 1.88(0.86, 4.11) | 0.001 |
| 4001–6000 | 31 | 79 | 1.39(0.74, 2.61) | 1.50(0.67, 3.37) | |
| 6001–7500 | 20 | 55 | 1.28(0.64, 2.58) | 1.99(0.82, 4.81) | |
| >7501 | 22 | 78 | 1.00 | 1.00 | |
| **Physical exercise** | | | | | |
| Poor | 97 | 172 | 4.02(2.21, 7.29) | 3.26(1.62, 6.53) | |
| Good | 15 | 107 | 1.00 | 1.00 | |
| **History of an eye examination** | | | | | 0.002 |
| Yes | 22 | 103 | 1.00 | 1.00 | |
| No | 90 | 176 | 2.39(1.41, 4.05) | 2.94(1.50, 5.76) | |
| **DM duration since diagnosis (year)** | | | | | |
| ≤3 | 20 | 80 | 1.00 | 1.00 | |
| 4–6 | 26 | 104 | 1.00(0.52, 1.91) | 0.87(0.41, 1.82) | 0.71 |
| 7–9 | 31 | 47 | 2.63(1.35, 5.14) | 4.53(1.99, 10.30) | 0.0001 |
| >9 | 35 | 48 | 2.91(1.51, 5.61) | 4.78(2.11, 10.83) | 0.0001 |
| **BMI(kg/m$^2$)** | | | | | 0.386 |
| Normal | 75 | 215 | 1.00 | 1.00 | |
| Underweight | 14 | 24 | 1.67(0.82 3.40) | 1.94(0.75, 5.01) | |
| Overweight and obese | 23 | 40 | 1.64(0.92, 2.93) | 1.13(0.56, 2.28) | |
| **Glycemic control** | | | | | 0.0001 |
| Good | 22 | 132 | 1.00 | 1.00 | |
| Poor | 90 | 147 | 3.67(2.17, 6.19) | 4.34(2.26, 8.34) | |
| **Hypertension** | | | | | 0.300 |
| Yes | 25 | 35 | 2.00(1.13, 3.53) | 1.43(0.72, 2.86) | |
| No | 87 | 244 | 1.00 | 1.00 | |
| **Peripheral neuropathy** | | | | | |
| Yes | 16 | 15 | 2.93(1.39, 6.16) | | 0.018 |
| No | 96 | 264 | 1.00 | 3.01(1.21, 7.50) | |

COR: crude odds ratio, AOR-adjusted odds ratio, CI: confidence interval

discrepancy might be due to the variation in the definition of visual impairment, study setting, and socio-demographic characteristics of study participants. For instance, visual impairment was defined as visual acuity less than 6/12 in either of the eyes in the study done in Dessie, while in this study, visual impairment was defined as visual acuity less than 6/12 in the better eye, which led to lowering the magnitude of visual impairment. Moreover, the study done in

Dessie included only type II diabetic patients who are considered to be older. This results in an increment of visual impairment due to the occurrence of age-related eye disease. Furthermore, the studies conducted in Gondar, Ethiopia, and Yemen were done on diabetic patients who were attending an eye care clinic for ophthalmic evaluation and already had compliant visual symptoms. Studying the prevalence in this setup would increase the magnitude of visual impairment. This finding indicted that stakeholders, particularly the minister of health will design a strategy on how to maintain an eye health through Telemedicine for diabetic patients. Evidence showed that screening of diabetic retinopathy using Telemedicine is very important to reduced diabetic related visual impairment [30, 31].

The result of this study was higher than the studies conducted in Malawi 5% [10], Zambia 17.1% [12], Nigeria 24.1% [32], Tunisia 22.2% [33], Ghana 18.4% [34], Jordan 17.5% [35], Turkey 13.5% [18], Armenia 13% [40], Sankara 4.1% [17] and South China 11.8% [22]. This difference might be the result of a variation in the definition and cut-off point of visual impairment. For example, the studies done in Malawi, Zambia, Nigeria, and Jordan defined visual impairment as having a visual acuity of less than 6/18, which resulted in a lowering of the magnitude of visual impairment in those studies as compared to this study. Furthermore, studies in Tunisia, Ghana, Turkey, and China reported the magnitude of visual impairment based on best corrected visual acuity, whereas this study used presenting visual acuity. Further-more, the availability and accessibility of eye care services and eye care service seeking behavior of the study participants could also contribute to this difference.

Participants who had diabetes for more than 7 years were 4.53 times more likely to develop visual impairment than those who had diabetes for ≤ 3 years. This result was consistent with the studies conducted in Dessie, Ethiopia [39], Tunisia [33], and Armenia [35]. The likely rea-son for this association is that prolonged diabetes can decrease insulin hormone production by the pancreas or result in target cell resistance. This, in turn, increases the risk of developing diabetic retinopathy and other diabetic eye complications that cause visual impairment [36].

The risk of developing visual impairment in those participants aged 60–80 years was 4.30 times higher than in participants aged 20–40 years, which was supported by the findings in Dessie, Ethiopia, Tunisia, Turkey, and India. The possible explanation for this association is that as age increases, the coexistence of age-related eye diseases such as degenerative maculo-pathy, cataracts, and glaucoma could increase. Those conditions can increase visual impairment among older diabetic patients [37].

The odds of visual impairment among participants who had poor physical exercise were 2.93 times greater than those who had good physical exercise, which was supported by a simi-lar study done in Dessie, Ethiopia [15]. This might be because poor physical exercise can impede the rapid enhancement of skeletal muscle oxidative capacity, insulin sensitivity, and glycemic control in diabetic patients, particularly patients with type 2 diabetes, resulting in worsening ocular complications [38, 39]. Besides, a wrong lifestyle may produce an excessively positive caloric balance, thus causing insulin resistance by enhancing visceral adipose tissue and consequently releasing a much higher level of free fatty acids, TNF-α, adipokines and hyperglycemia which will lead to increased inflammation, endothelial dysfunction and, as a consequence to increased prevalence of diabetes complications and in particular retinopathy [40].

Participants with poor glycemic control were 2.61 times more likely to have visual impairment than those with good glycemic control. This result was similar to that of the study conducted in Dessie, Ethiopia. The possible justification for this association is that an incre-ment in the level of hyperglycemia or having poor glycemic control can increase the onset and rate of progression of diabetic retinopathy, which leads to visual impairment [19, 41].

Participants who had never visited an eye care clinic were 2.43 times more likely to have visual impairment than those participants who had a history of eye examinations. This might be due to the fact that the utilization of eye care services for diabetic patients is crucial for handling sight-threatening diabetic eye complications. In contrast, those people who do not have a history of eye examinations are highly susceptible to undiagnosed diabetic eye disease.

Participants with peripheral neuropathy were 3.01 times more vulnerable to developing visual impairment as compared to participants without peripheral neuropathy. This result is similar to the studies done in Tunisia and Sankara, India [17, 33]. The plausible reason for this association might be due to diffuse retinal neurodegenerative changes in diabetic patients prior to retinopathy development, and visual sensitivity decreases disproportionately with increasing eccentricity in diabetic patients with peripheral neuropathy, resulting in loss of vision [42, 43].

## Limitation of the study

Due to the cross-sectional nature of the study, it could not show a cause-effect relationship. Since the study was done in a single center it does not represent real burden of visual impairment in a general diabetic population. In addition, fasting blood sugar was used to assess glycemic control, due to the lack of facilities to assess glycated hemoglobin in the study area. Moreover, there is also a lack of data on albuminuric chronic kidney disease and HDL levels, which were significantly associated with diabetic retinopathy in previous studies that affect the magnitude of visual impairment.

## Conclusion

The study found that high prevalence of visual impairment among adult diabetic patients. Older age, longer duration of diabetes, poor physical exercise, poor glycemic control, history of eye examination, and peripheral neuropathy were significantly associated with visual impairment. Diabetic patients should engage in regular physical activity for at least 150 minutes per week, control their glucose levels with proper use of medication, and have regular eye exams.

## Author Contributions

**Conceptualization:** Henok Biruk Alemayehu, Mikias Mered Tilahun.

**Data curation:** Melkamu Temeselew Tegegn, Mikias Mered Tilahun.

**Formal analysis:** Henok Biruk Alemayehu, Melkamu Temeselew Tegegn.

**Funding acquisition:** Melkamu Temeselew Tegegn.

**Investigation:** Henok Biruk Alemayehu.

**Methodology:** Henok Biruk Alemayehu, Melkamu Temeselew Tegegn.

**Project administration:** Mikias Mered Tilahun.

**Resources:** Henok Biruk Alemayehu.

**Software:** Melkamu Temeselew Tegegn.

**Supervision:** Melkamu Temeselew Tegegn, Mikias Mered Tilahun.

**Validation:** Mikias Mered Tilahun.

**Visualization:** Henok Biruk Alemayehu, Melkamu Temeselew Tegegn.

**Writing – original draft:** Henok Biruk Alemayehu.

**Writing – review & editing:** Mikias Mered Tilahun.

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
