## [Decision Letter · Decision Letter 0]

12 Sep 2022

PONE-D-22-24140Prevalence and Associated Factors of Visual Impairment Among Adult Diabetic Patients Visiting Adare General Hospital, Hawassa, South Ethiopia, 2022PLOS ONE

Dear Dr. Tilahun,

Thank you for submitting your manuscript to PLOS ONE. After careful consideration, we feel that it has merit but does not fully meet PLOS ONE’s publication criteria as it currently stands. Therefore, we invite you to submit a revised version of the manuscript that addresses the points raised during the review process.

ACADEMIC EDITOR:

Please submit your revised manuscript by Oct 27 2022 11:59PM. If you will need more time than this to complete your revisions, please reply to this message or contact the journal office at plosone@plos.org. Please include the following items when submitting your revised manuscript:A rebuttal letter that responds to each point raised by the academic editor and reviewer(s). You should upload this letter as a separate file labeled 'Response to Reviewers'.A marked-up copy of your manuscript that highlights changes made to the original version. You should upload this as a separate file labeled 'Revised Manuscript with Track Changes'.An unmarked version of your revised paper without tracked changes. You should upload this as a separate file labeled 'Manuscript'.

We look forward to receiving your revised manuscript.

Kind regards,

Ferdinando Carlo Sasso, PhD, MD

Academic Editor

PLOS ONE

Journal Requirements:

Additional Editor Comments:

The reviewers raised several issues about discussion section and study limitations.

Please, address all issues raised by the reviewers and submit a revised manuscript.

Reviewers' comments:

Reviewer's Responses to Questions

**Comments to the Author**

1. Is the manuscript technically sound, and do the data support the conclusions?

Reviewer #1: Partly

Reviewer #2: Yes

2. Has the statistical analysis been performed appropriately and rigorously? 

Reviewer #1: Yes

Reviewer #2: Yes

3. Have the authors made all data underlying the findings in their manuscript fully available?

Reviewer #1: Yes

Reviewer #2: Yes

4. Is the manuscript presented in an intelligible fashion and written in standard English?

Reviewer #1: Yes

Reviewer #2: Yes

5. Review Comments to the Author

Reviewer #1: I read with great interest the paper “Prevalence and Associated Factors of Visual Impairment Among Adult Diabetic Patients Visiting Adare General Hospital, Hawassa, South Ethiopia, 2022" by Mikias Mered Tilahun et al.

The article is well written. Paper design is fine. The article is logically divided into sections and subsections. The analysis is well performed

Comments:

1. Limitation of the study is also represented by the lack of data about 2 major risk factors that have been associated to diabetic retinopathy, that are albuminuric chronic kidney disease and high HDL levels, which should be added to the appropriate section.

2. Discussion should be treated more in depth. For example, the role played by physical activity in ameliorating insulin resistance. In fact, a wrong lifestyle may produce an excessively positive caloric balance, thus causing insulin resistance by enhancing visceral adipose tissue and consequently releasing a much higher level of free fatty acids, TNF-α, adipokines and hyperglycaemia which will lead to increased inflammation, endothelial disfunction and, as a consequence to increased prevalence of diabetes complications and in particular retinopathy (doi: 10.3390/antiox10020270).

Reviewer #2: Dear Editor

I’ve read with great interest the manuscript draft “Prevalence and Associated Factors of Visual Impairment Among Adult Diabetic Patients Visiting Adare General Hospital, Hawassa, South Ethiopia, 2022” by Henok Biruk Alemayehu et al. However, some issues need to be raised.

Abstract:

- There are some typing errors.

Introduction:

- “Diabetes is a complex…” seems too generic, I would suggest to specify which type of diabetes.

- Please insert the entire words when an acronym is used for the first time throughout the text.

Methods:

- The authors write “systemic co-morbidities were collected from each patient's medical chart”. I would suggest to briefly specify how co-morbidities diagnosis has been done, particularly for what concerns the statistically associated peripheral neuropathy.

Discussion:

- The authors write “Studying the prevalence in this setup would increase the magnitude of visual impairment.” Recently, some authors have demonstrated that telemedicine could allow to implement the screening programs also during a pandemic period (Galiero R, Pafundi PC, Nevola R, et al. The Importance of Telemedicine during COVID-19 Pandemic: A Focus on Diabetic Retinopathy. J Diabetes Res. 2020;2020:9036847. Published 2020 Oct 14. doi:10.1155/2020/9036847; Sasso FC, Pafundi PC, Gelso A, et al. Telemedicine for screening diabetic retinopathy: The NO BLIND Italian multicenter study. Diabetes Metab Res Rev. 2019;35(3):e3113. doi:10.1002/dmrr.3113). I would suggest to briefly discuss this point.

Limitations:

- This is a single-center study, thus it seems difficult to fully generalize the results. I would suggest to add this issue to limitations section.

- To evaluate glycemic control, the study lacks glycated hemoglobin, I would suggest to discuss this issue or to move in to limitation section.

Conclusion:

- The authors write “Diabetic patients should engage in regular physical activity for at least 150 minutes per week, control their glucose levels with proper use of medication, and have regular eye

exams.”, however no references have been associated with this sentence. I would suggest to at least a reference linked to this point.

6. PLOS authors have the option to publish the peer review history of their article (what does this mean?). If published, this will include your full peer review and any attached files.

Reviewer #1: No

Reviewer #2: No

---

## [Author Response · Author response to Decision Letter 0]

29 Sep 2022

Point by point response to reviewer`s comment

Manuscript title: Prevalence and Associated Factors of Visual Impairment Among Adult Diabetic Patients Visiting Adare General Hospital, Hawassa, South Ethiopia, 2022

Manuscript number: 

Dear Reviewer. Thank you for providing us the chance to revise our manuscript. We are great full for your careful and in-depth reading of the manuscript. We appreciate your thoughtful comments and constructive suggestions, which help us to improve the quality of the manuscript. Having this we addressed all the concerns raised and incorporated our reflections in the revised manuscript. We attempted to address all the issues raised to be corrected. 

Many thanks

 Responses to Reviewers 

Reviewer- 1

1. Reviewer’s comment: Limitation of the study is also represented by the lack of data about 2 major risk factors that have been associated to diabetic retinopathy, that are albuminuric chronic kidney disease and high HDL levels, which should be added to the appropriate section.

Authors’ response: Thank you for your comment. We accept the comment and made necessary correction in the revised manuscript. 

2. Reviewer’s comment: Discussion should be treated more in depth. For example, the role played by physical activity in ameliorating insulin resistance. In fact, a wrong lifestyle may produce an excessively positive caloric balance, thus causing insulin resistance by enhancing visceral adipose tissue and consequently releasing a much higher level of free fatty acids, TNF-α, adipokines and hyperglycaemia which will lead to increased inflammation, endothelial disfunction and, as a consequence to increased prevalence of diabetes complications and in particular retinopathy (doi: 10.3390/antiox10020270).

Authors’ response: Thank you for your depth insight. We accept it and made incorporated this relevant information with in the revised manuscript. 

 Reviewer - 2

1. Reviewer’s comment: Abstract: There are some typing errors.

Authors’ response: Thank you for your comment. We accept the comment and made necessary correction in the revised manuscript

2. Reviewer’s comment: Introduction: “Diabetes is a complex…” seems too generic, I would suggest to specify which type of diabetes.

Authors’ response: Thank you for your suggestion. We accept it and we tried to address your suggestion in the revised manuscript 

But diabetes both type I and II were found resulted hyperglycemia with different mechanism. 

3. Reviewer’s comment - Please insert the entire words when an acronym is used for the first time throughout the text

Author response: Thank you for your comment. We made correction within the revised manuscript 

4. Reviewer’s comment: Methods - The authors write “systemic co-morbidities were collected from each patient's medical chart”. I would suggest to briefly specify how co-morbidities diagnosis has been done, particularly for what concerns the statistically associated peripheral neuropathy.

Author response: Thank for your comment. We accept the insight and include the clinical diagnostic criteria of peripheral neuropathy in the revised manuscript. Even though here were the diagnosis criteria of other systemic co morbidities included in our study Diagnosis of hypertension was made based on patients’ blood pressure when a person's systolic blood pressure is ≥140 mm Hg and/or their diastolic blood pressure ≥90 mm Hg following repeated examination. Anemia was diagnosed based on laboratory results of blood hemoglobin concentration (below 130 g/L for men, 120 g/L for females). Heart disease and nephropathy were diagnosed by their respective clinics, and patients were already on their medication. Peripheral neuropathy was diagnosed based on a person’s medical history and physical examination (inspection of feet, vibration perception, ankle reflex test, and monofilament test)

5. Reviewer’s comment: Discussion: The authors write “Studying the prevalence in this setup would increase the magnitude of visual impairment.” Recently, some authors have demonstrated that telemedicine could allow to implement the screening programs also during a pandemic period (Galiero R, Pafundi PC, Nevola R, et al. The Importance of Telemedicine during COVID-19 Pandemic: A Focus on Diabetic Retinopathy. J Diabetes Res. 2020;2020:9036847. Published 2020 Oct 14. doi:10.1155/2020/9036847; Sasso FC, Pafundi PC, Gelso A, et al. Telemedicine for screening diabetic retinopathy: The NO BLIND Italian multicenter study. Diabetes Metab Res Rev. 2019;35(3):e3113. doi:10.1002/dmrr.3113). I would suggest to briefly discuss this point.

Author response: Thank you for your suggestion. We accept it and we tried to show the impact of Telemedicine for screening diabetic retinopathy on reduction of visual impairment among patients with diabetes. 

6. Reviewer’s comment: Limitations - This is a single-center study, thus it seems difficult to fully generalize the results. I would suggest adding this issue to limitations section. 

- To evaluate glycemic control, the study lacks glycated hemoglobin, I would suggest to discuss this issue or to move in to limitation section.

Authors’ response: Thank you for your comment. We accept the comment and made necessary correction in the new revised manuscript. 

7. Reviewer’s comment: Conclusion:

- The authors write “Diabetic patients should engage in regular physical activity for at least 150 minutes per week, control their glucose levels with proper use of medication, and have regular eye exams.”, however no references have been associated with this sentence. I would suggest to at least a reference linked to this point

Author response: Thank you for your critical insight. But in this study, poor physical exercise, poor glucose control and having no history of eye examination were significantly associated with visual impairment. Based on this fining, we have recommend that those patient with diabetes should engage in regular physical activity for at least 150 minutes per week, control their glucose levels with proper use of medication, and have regular eye exams. So, we believed that no need of referencing.

---

## [Decision Letter · Decision Letter 1]

2 Oct 2022

Prevalence and Associated Factors of Visual Impairment Among Adult Diabetic Patients Visiting Adare General Hospital, Hawassa, South Ethiopia, 2022

PONE-D-22-24140R1

Dear Dr. Tilahun,

We’re pleased to inform you that your manuscript has been judged scientifically suitable for publication and will be formally accepted for publication once it meets all outstanding technical requirements.

Kind regards,

Ferdinando Carlo Sasso, PhD, MD

Academic Editor

PLOS ONE

Additional Editor Comments (optional):

The authors addressed all issues raised by reviewers.

Reviewers' comments:

Reviewer's Responses to Questions

**Comments to the Author**

1. If the authors have adequately addressed your comments raised in a previous round of review and you feel that this manuscript is now acceptable for publication, you may indicate that here to bypass the “Comments to the Author” section, enter your conflict of interest statement in the “Confidential to Editor” section, and submit your "Accept" recommendation.

Reviewer #1: All comments have been addressed

Reviewer #2: All comments have been addressed

2. Is the manuscript technically sound, and do the data support the conclusions?

Reviewer #1: Yes

Reviewer #2: Yes

3. Has the statistical analysis been performed appropriately and rigorously? 

Reviewer #1: Yes

Reviewer #2: Yes

4. Have the authors made all data underlying the findings in their manuscript fully available?

Reviewer #1: Yes

Reviewer #2: Yes

5. Is the manuscript presented in an intelligible fashion and written in standard English?

Reviewer #1: Yes

Reviewer #2: Yes

6. Review Comments to the Author

Reviewer #1: The authors answered appropriately to all my quarries. The paper can now be further processed for publication

Reviewer #2: (No Response)

7. PLOS authors have the option to publish the peer review history of their article (what does this mean?). If published, this will include your full peer review and any attached files.

Reviewer #1: No

Reviewer #2: No

---

## [Editor Report · Acceptance letter]

5 Oct 2022

PONE-D-22-24140R1 

Prevalence and Associated Factors of Visual Impairment Among Adult Diabetic Patients Visiting Adare General Hospital, Hawassa, South Ethiopia, 2022 

Dear Dr. Tilahun:

I'm pleased to inform you that your manuscript has been deemed suitable for publication in PLOS ONE. Congratulations! Your manuscript is now with our production department. 

Kind regards, 

on behalf of

Professor Ferdinando Carlo Sasso 

Academic Editor

PLOS ONE